# Research on 3D-Print Design Method of Spatial Node Topology Optimization Based on Improved Material Interpolation

**DOI:** 10.3390/ma15113874

**Published:** 2022-05-29

**Authors:** Xianjie Wang, Fan Zhang, Yang Zhao, Zhaoyi Wang, Guangen Zhou

**Affiliations:** 1Key Laboratory of Civil Engineering Structure and Mechanics, Inner Mongolia University of Technology, Hohhot 010051, China; xianjiewang@ynu.edu.cn; 2School of Architecture and Planning, Yunnan University, Kunming 650106, China; 18608581557@163.com; 3School of Civil Engineering, Zhejiang University, Hangzhou 310058, China; ceyzhao@zju.edu.cn; 4Zhejiang Southest Space Frame Co., Ltd., Hangzhou 310058, China; zgg1967@163.com

**Keywords:** improved BESO algorithm, optimal design of space nodes, material interpolation method, additive manufacturing, model postprocessing

## Abstract

Designing a high-strength node is significant for space structures. Topological optimization can optimally allocate the material distribution of components to meet performance requirements. Although the material distribution after topology optimization is optimum, the structure becomes complicated to manufacture. By using additive manufacturing technology, this problem can be well solved. At present, both topology optimization technology and additive manufacturing technology are quite mature, but their application in the design of spatial nodes is very recent and less researched. This paper involves the study and improvement of the node optimization design–manufacturing integrated method. This study used the BESO optimization algorithm as the research algorithm. Through a reasonable improvement of the material interpolation method, the algorithm’s dependence on the experience of selecting the material penalty index P was reduced. On this basis, the secondary development was carried out, and a multisoftware integration was carried out for optimization and manufacturing. The spatial node was taken as the research object, and the calculation results of the commercial finite element software were compared. The comparison showed that the algorithm used in this paper was better. Not only was it not trapped in a local optimum, but the maximum stress was also lower. In addition, this paper proposed a practical finite element geometric model extraction method and smoothing of the optimized nodes, completing the experiment of the additive manufacturing forming of the nodes. It provides ideas for processing jagged edges brought by the BESO algorithm. This paper verified the feasibility of the multisoftware integration method of optimized manufacturing.

## 1. Introduction

The evolutionary structural optimization (ESO) method and its improved method (BESO) were first proposed by Xie [1,2] et al. The method gradually removes the inefficient materials in a structure so that the structure can gradually reach the optimal state in terms of performance. At the same time, this method has an extensive prospect for secondary development. With the continuous advancement of computer technology and 3D-printing technology, topology optimization is no longer satisfied with just finding the shape, but has turned its attention to new technologies that are integrated with additive manufacturing. Regarding topology optimization techniques for additive manufacturing, there has been a large amount of research beneficial to practical applications. First, in the study of manufacturing molding, the researchers considered the process constraints of additive manufacturing technology, including self-supporting constraints [3], maximum and minimum constraints [4,5], and anisotropic constraints [6,7]. This type of research is key to the fusion of topology optimization and additive manufacturing. The second is the defects and properties of additive manufacturing materials and components [8]. The researchers studied the anisotropy of additive manufacturing components. It turned out that this anisotropy is ubiquitous, not only for metal materials [9], but also for polymer materials [10,11]. Although additive manufacturing encounters common manufacturing precision errors, the mechanical properties of additively manufactured products can be well matched in experiments and finite element analysis [12], which is beneficial to obtaining accurate and stable material properties for topology optimization analysis. According to statistics [13], there are currently more than 200 additive manufacturing technologies worldwide. Among the many fields of additive manufacturing services, the research on additive manufacturing in the construction field began late. The reason is that, firstly, the design of various structures has specific specifications, and secondly, the current commercial topology optimization software is not open enough to meet the customization requirements. In space-structure design, because space nodes require a high strength and a light weight, they have great potential in optimized manufacturing, which has aroused the high interest of researchers. Existing studies include bifurcated nodes [14], rod-tie nodes [15], cable dome nodes [16], and aluminum alloy nodes for exterior wall facades [17]. The fusion of topology optimization and additive manufacturing makes the design of space nodes more accessible and convenient, especially complex models after topology optimization, which can also be easily manufactured. These studies were all based on using the SIMP method, and the current BESO method still has some problems in practical application: less software and the existence of rough edges after optimization. Based on MATLAB and Abaqus, this paper carries out the secondary development of the BESO method, discusses the further improvement of the material method with an aim to establish a convenient and free integrated method, and then provides ideas for the optimal manufacturing of space nodes.

This study took space nodes with complex forces as the research object. The following work was done using the combination of 3D printing and topology optimization. First, the traditional material interpolation method was improved. The improved material interpolation formula could automatically penalize according to the material’s properties. Secondly, based on the secondary development of ABAQUS, a platform for the integration process of structure from optimization to manufacturing was designed by using MATLAB, and the molding experiment was carried out. Finally, the optimization results of this study were compared with the results of the Tosca module that is supplied with ABAQUS. By comparison, it was found that the optimization results reported this paper were better than the results of Tosca in terms of the mechanical properties. When the percentage of volume optimization was low, the shape of the optimization results found in this study were more reasonable.

## 2. BESO Topology Optimization Method

### 2.1. Mathematical Description of Topology Optimization

This article only discusses the topology optimization of space nodes under static loads. Taking the minimum structural flexibility as the objective function, the strain energy density *X_i_* of the element as the design variable, and the volume fraction as the constraint condition, the optimization problem can usually be described as:(1)$$\left\{\begin{array}{ll} min & C=\frac{1}{2}F^{T}U\\ s.t. & V^{*}-\sum_{i}^{n}V_{i}=0 \end{array}\right.$$

In the formula, ***F***^T^ represents the transposition of the load matrix, ***U*** represents the displacement matrix, *V** is the target volume fraction, *V_i_* is the volume of the *i_th_* unit, *X_i_* is the strain energy density of the *i_th_* unit, and n is the total number of structural units. It can be equivalent to:(2)$$\left\{\begin{array}{ll} min & C=\sum_{i=1}^{n}V_{i}X_{i}\\ s.t. & V^{*}-\sum_{i=1}^{n}V_{i}=0 \end{array}\right.$$

Under the condition of satisfying the maximum structural rigidity, the volume fraction will be continuously reduced until it is reduced to the target volume fraction.

### 2.2. Improved Material Interpolation Method

The current commonly used material interpolation method is an interpolation method widely used in the variable density method [18,19]. This method can tend the density of the unit to 0/1 and make the density division of the unit continuous. When the BESO method uses the material interpolation model [20,21], in order to make the division of the element clearer, it is often necessary to define a very small soft element parameter *X_min_* to make it infinitely close to 0 to replace the void area after the element is deleted [22]. This is to prevent the sensitivity of the soft cell from being directly equal to 0 and causing a matrix singularity.

In the past, the penalty index value that needed to be selected was 1.8 in order to produce practical effects. At the same time, it was affected by the penalty index and the artificially selected minimum value, as well as the two material properties. Not only was the efficiency low, but it also had many parameters and was greatly affected by humans.

In order to improve this problem and improve the adaptability of the algorithm, we made necessary modifications to it:(3)$$E\left(X_{i}\right)={\left(E_{1}-E_{2}\right)}^{\wedge }\left(1-p\right)+E_{2}$$
where *p* is the penalty index of the material, *E*_1_ is the elastic modulus of the solid element material, and *E*_2_ is the elastic modulus of the soft element material. Then, the material interpolation method is:(4)$$a_{i}=\left\{\begin{array}{ll} \frac{1}{2}\left(1-\frac{E_{2}}{E_{1}}\right)U^{T}KU & \;\mathrm{for}\;\mathrm{solid}\;\mathrm{element}\;\mathrm{material}1\\ \frac{1}{2}\left(\frac{{\left(E_{1}-E_{2}\right)}^{\wedge }\left(1-P\right)}{{\left(E_{1}-E_{2}\right)}^{\wedge }\left(1-P\right)+E_{2}}\right)U^{T}KU & \mathrm{for}\;\mathrm{soft}\;\mathrm{element}\;\mathrm{material}2 \end{array}\right.$$
where a*_i_* is the sensitivity of the element, *K* is the stiffness matrix calculated from the elastic moduli *E*_1_ and *E*_2_, and *U* is the displacement matrix.

When *E*_1_ > *E*_2_ and the difference is large, the above formula can be equivalent to:
(5)$$a_{i}=\left\{\begin{array}{ll} \frac{1}{2}\left(1-\frac{E_{2}}{E_{1}}\right)U^{T}KU & \mathrm{for}\;\mathrm{solid}\;\mathrm{element}\;\mathrm{material}1\\ \frac{1}{2}\left(\frac{0}{0+E_{2}}\right)U^{T}KU & \;\mathrm{for}\;\mathrm{soft}\;\mathrm{element}\;\mathrm{material}2 \end{array}\right.$$
which is:(6)$$a_{i}=\left\{\begin{array}{cc} \frac{1}{2}\left(1-\frac{E_{2}}{E_{1}}\right)U^{T}KU & \mathrm{for}\;\mathrm{solid}\;\mathrm{element}\;\mathrm{material}1\\ \;0 & \;\mathrm{for}\;\mathrm{soft}\;\mathrm{element}\;\mathrm{material}2 \end{array}\right.$$

As shown in Figure 1, through the adjustment formula, when the penalty index *p* started from 0, the soft cell sensitivity was in a reduced state.

As the penalty index increased, the numerator of the material interpolation formula shrank, so the result of the formula also shrank. By multiplying the calculated value of the soft element obtained by the finite element calculation, the sensitivity value of the soft element after the penalty was obtained, which gave practical meaning to the soft element as the deleted part in the calculation. In addition, in the following sensitivity filtering, the use of this method could also reduce the contribution of the soft unit sensitivity.

Therefore, the improved method played the following roles:

To a certain extent, the optimization instability caused by the artificial selection of the *p*-value was weakened. At the same time, the enhanced penalty effect can be seen in Figure 1; when *p* was greater than 0.5, the sensitivity was reduced, and the penalty effect had begun to occur.

The artificially selected minimum value *X_min_* was deleted, and the formula result was changed to be automatically adjusted according to the elastic modulus of the solid element and soft element materials, which increased the adaptability of the material interpolation formula.

### 2.3. Optimization Criteria

The deletion and retention criteria of the unit generally adopt the dichotomy to set the sensitivity threshold of the unit division, which can be defined as:(7)$$\left\{\begin{array}{ll} V_{th}=\frac{1}{2}(a_{max}+a_{min})\\ V_{th}=\frac{1}{2}(a_{max}+th) & \begin{array}{l} \;V_{add}>V^{*} \end{array}\\ V_{th}=\frac{1}{2}(th+a_{nin}) & V_{del}<V^{*} \end{array}\right.$$

In the formula, *V_th_* is the sensitivity threshold that determines whether the cell is deleted or retained, *a_max_* is the maximum sensitivity, *a_min_* is the minimum sensitivity, *V_add_* is the volume of the solid unit, *V_del_* is the volume of the soft unit, and *V** is the target volume.

In order to realize the evolution process of the improved algorithm, the initial sensitivity threshold is defined as *V_th_* = (*a_max_* + *a_min_*)/2.

During the iteration process, we made the following mathematical description:

When *V_add_* > *V**, then *a_min_* = (*a_min_* + *V_th_*)/2; *V_th_* = (*a_min_* +*a_max_*)/2, when *V_add_* < *V**, then *a_max_* = (*V_th_* + *a_max_*)/2; *V_th_* = (*a_min_* + *a_max_*)/2.

This continued until the goal was achieved; that is, when *V_add_* = *V**, the iteration ended.

### 2.4. Optimization of the Process

When performing the analysis of the topology optimization solution of the 3D model, the steps were as follows:
(1)Firstly, establish the ABAQUS finite element model that needs to be optimized, and output the finite element calculation model INP file.(2)ABAQUS is called through the MATLAB background to perform the finite element analysis. After the calculation and analysis are completed, the ODB database function of Python is used to extract all the element design variables X*i* of the solved design domain.(3)Then, the sensitivity analysis based on the improved material interpolation method is performed on the extracted data in MATLAB, and then the optimization analysis is performed based on the optimization criterion.(4)The elements that need to be deleted and retained are divided into different element sets, written as a new finite element model INP file, and iterated again until the target volume fraction is reached and the optimization ends.(5)After the optimization, extract the 3D model format file of the finite element model composed of the retained element set, and perform the additive manufacturing.

Throughout the process, RHINO, ABAQUS, MATLAB, and Python were used for optimization, smoothing, and then 3D printing using the printer’s supporting software.

The process is shown in Figure 2 below.

## 3. Space Node Optimization-Manufacturing Example

### 3.1. Model Parameters and Modeling

There are various forms of space nodes, and the combination of node design and topology optimization makes the diversity of node shapes fully developed. The designer can optimize the established model according to the connection method of the nodes, load application, and boundary conditions, etc., and obtain the topological results freely generated based on actual needs after optimization. Therefore, in this design area, a sufficient design space needs to be provided.

In the 3D modeling software Rhino, we built a square node model [23] with ear plates as shown in Figure 3.

The model parameters are given in Table 1.

We imported the established model into the finite element software in a 3D model file format, added attributes to it, and divided the mesh. Figure 4 shows the finite element model with meshes. The blue part in the figure is the design domain, and the red part is the connecting part of the ear plate. The grid size of the design domain was 2 mm and the unit was C3D8R, with a total of 64,000 units.

Considering that the bending moment had little influence on the actual project, the pure shear force, the pure axial force, and the combined force of the combination of the two were mainly considered in the optimization. Six degrees of freedom constraints were imposed on the end face of the connecting part to place one end in a fixed state. Figure 5 is a schematic diagram of the working conditions of applying a shear load, axial load, and combined load.

In the figure, the blue area on the end face is the fixed end of the boundary; the pink load is the axial force, which was uniformly applied to the end face of the ear plate in the form of surface load; and the yellow arrow is the shear force, which was applied to the end face of the ear plate in the form of a point load.

### 3.2. Optimization Effect before and after Improvement

The optimization results need to consider the influence of boundary burrs and checkerboards in actual applications. The improved material interpolation method greatly weakened the impact of the penalty index *p*, and the penalty effect was more dependent on the gap between the material properties. When the material properties of the soft element and the solid element were quite different in magnitude, the sensitivity was very adaptive, and it had a good performance in the control of the checkerboard. The sensitivity could be adjusted to the infinitesimal range according to the difference between the materials; e.g., the combined load case. As shown in Figure 6, the optimized structure had a good printable shape.

The material’s elastic modulus and target volume fraction were kept constant with no checkerboard control. As shown in Figure 7, because the unit stress became concentrated and complicated near the connection position of the ear plate, it was prone to unclear unit selection. Therefore, it can be seen that there was a local checkerboard effect at the position where the ear plate was connected to the node.

Compared with the optimization results in Figure 7, the improved optimization results in Figure 6 was obviously more printable.

### 3.3. Analysis of Optimization Results under Different Working Conditions

#### 3.3.1. Axial Force Acting Conditions

In order to make the optimization results clearer, the target volume fraction was set very low. Taking a volume fraction of 20% as an example, the evolution rate ER was 0.08. The objective function of the structural optimization was the maximum stiffness.

Figure 8a,b show graphs of the optimization results under pure axial force conditions. It can be seen that the optimized structure topology was clear, and the unit area between adjacent lugs was curved under the action of axial force, which conformed to its stress condition. There was no checkerboard phenomenon as a whole, and the connectivity was good.

As shown in Figure 8b, there was a uniform higher stress area between adjacent ear plates, while the middle part of the node had a lower stress value, and the overall force was uniform. This meant that the structure as a whole was in a state of high-efficiency stress, which was in line with the actual stress situation. Figure 9 shows the change in total compliance versus the volume fraction for the overall optimization process. With the continuous reduction in the volume fraction, the total compliance increased, indicating that the remaining material was used efficiently. When the volume fraction reached the target value, the total compliance fluctuated continuously, and eventually converged.

#### 3.3.2. Shear Conditions

Similarly, when the target volume fraction was set to 20%, the evolution rate was 0.08, the optimization target was the maximum stiffness, and the shear condition analysis was performed.

Figure 10a,b show the stress nephograms of the optimization results. Under the action of shear, the node can be approximately regarded as the fusion of three cantilever beams. As shown in Figure 10a, an X-shaped support structure appeared in the middle of the node, which conformed to the topology of the cantilever beam under shear force. The optimization results had no checkerboard phenomenon as a whole.

From the point of view of the stress distribution, the maximum value appeared at the connection between the node and the ear plate. After optimization, the overall stress distribution of the structure was uniform, indicating that the overall structure was in a highly efficient state of stress. In Figure 11, as the volume fraction continued to decrease, the average flexibility increased until the target volume fraction was reached, and the flexibility value began to converge. This purpose of this stage was to continuously optimize the element distribution without changing the target volume. The flexibility value gradually converged accurately, and finally, the optimized optimization results were obtained.

#### 3.3.3. Combined Load Conditions

In the same way, the combined working condition analysis was carried out.

Figure 12 and Figure 13 show the optimization results and the optimization process under the combined load, respectively. Figure 12a shows that the force transfer paths of the whole structure were effectively utilized, and Figure 12b shows that the stress distribution was uniform. The modeling conformed to the combination of the optimization results under the two single working conditions of axial force and shear force. The upper part of the node was affected by the axial force to maintain a closed diamond top surface, and under the action of the shear force, the energy of the upper element was strengthened and the energy of the lower element was weakened, so the upward arc-shaped element accumulation appeared at the end of the boundary constraint.

### 3.4. Algorithm Comparison

In order to compare and discuss the advantages and disadvantages of the BESO algorithm, the optimized result of the SIMP algorithm was selected as the comparison object here. As one of the current mainstream algorithms, the SIMP algorithm is embedded in it as an optimization tool in many finite element software programs. The Tosca module that comes with ABAQUS uses the SIMP algorithm as an optimization solution tool. In order to ensure the reliability of the comparison, this section substituted the entire node model above into ABAQUS to solve.

The comparison used nodes under shear conditions and complex combined conditions as the comparison object. The objective function set in ABAQUS was still the minimum strain energy, the volume was used as the constraint condition, and the value was 20%.

As shown in Figure 14, when the SIMP algorithm was used for optimization, the stress of the overall structure under the shear condition was uniform, the area of maximum stress was concentrated in the middle of the thin rod, and there was a clear difference. As shown in Table 2, the maximum stress of the SIMP algorithm results is 10.12% larger than the BESO algorithm results, and the minimum stress is smaller than the BESO results. It should be noted that when using SIMP to optimize, the volume in the first step was reduced to 20%. The stress distribution of the BESO optimization results was more uniform, indicating that the whole structure was in a highly efficient load state, which effectively reduced the damage caused by the stress concentration in the structure. The circle in Figure 15 is the unit that was deleted by mistake in the first step, and the unit was not supplemented even in subsequent iterations. In the calculation of BESO, compared with SIMP’s volume-shrinkage method, its method of gradually reducing the volume ensured that the weak force-transmitting members of the structure could be retained.

As shown in Table 3, the overall stress of the results obtained by the SIMP algorithm appeared to be lower, but the optimization results of the SIMP algorithm were still far from the BESO algorithm, and the maximum stress was 25.48% higher. Comparing Figure 16 and Figure 17, it can be seen that SIMP described the skeleton of the structure more three-dimensionally, and the appearance was smoother. The BESO algorithm used the small squares of each unit of the structure as the carrier of design variables, so the optimization results also used the small squares as the structural characterization unit. Although BESO showed a more excellent performance, it still needs to be smoothed before printing and manufacturing.

## 4. Optimizing Results for Additive Manufacturing

### 4.1. Model Postprocessing

When using ABAQUS finite element analysis software, under normal circumstances, it is impossible to modify the geometric features of the model in the generated INP file. In this paper, a geometric model extraction method based on the finite element model of the element set partition was innovated, and the optimized element set geometric model extracted the OBJ format.

When the BESO algorithm was combined with the finite element software, in order to ensure the two-way addition and deletion of elements, the deleted and retained elements were divided into two material sets: set-1 and set-2, respectively. Therefore, the part of the unit set that needed to be deleted was still retained in the final optimization result model. The method used in this paper needed to extract the optimized results from the INP file after optimization.

The model extraction method is defined as follows:(8)$$\left\{\begin{array}{l} M=M_{1}\cap M_{e}\\ N=N_{1}\cap N_{2}\\ N^{’}=N\cap N_{2} \end{array}\right.$$

In the formula, ***M***_1_ is the element number matrix of the first material set set-1. ***M***_e_ is the matrix of all elements and their associated node number. Through the intersection of ***M***_1_ and ***M***_e_, the matrix ***M*** containing the unit number of the first material set and the number of its associated nodes is obtained. ***N*** is the matrix of the intersection of the node number of the first material element and the node number of the second material element, ***N***_1_ is the node number of the (set-1) solid element unit, and ***N***_2_ is the node number of the soft element (set-2). ***N***′ is the sequence number matrix of all nodes that are not related to the entity unit.

The BESO method was realized through the collaboration of MATLAB and ABAQUS. However, the optimized model obtained by the BESO method at this time was a model composed of unit blocks, and the model had a jagged boundary, which is not friendly to printing. In order to obtain a smooth boundary, the smoothing function of the three-dimensional modeling software Rhino was used to adjust it, and the boundary smoothing process was performed by adjusting parameters such as the smoothing coefficient and the amount of smoothing. Examples of the optimization results under the combined load condition with checkerboard control are shown in Figure 18 and Figure 19.

All the jagged boundaries of the model were smoothed, and the model was in a printable state as a whole.

### 4.2. 3D-Printing Effect

Regarding topological results, Yang Zhao et al. [24] printed and manufactured cable and rod nodes, and Longxuan Wang et al. [25] printed and manufactured four-forked steel casting nodes. Both used a method of combining lost wax casting, which was a more economical, practical method of manufacturing. In this method, a node mold was manufactured by printing a wax mold, and then the node was manufactured by casting.

In this study, related forming experiments were carried out using the optimized results. The STL format file of the smoothed model was exported from Rhino and loaded into the software supporting the additive manufacturing machine for adjustment of the printing process parameters. The support methods were usually column-shaped support and tree-shaped support. Compared with column-shaped support, the tree-shaped support can save material and provide stronger support. In this study, some 4 mm diameter tree supports were added to support the printing of some suspended parts. The printing process used in this study was the FDM fused deposition manufacturing printing technology. In the research on the properties of PLA-printed parts, Sohrabian and Vaseghi [12] et al. carried out detailed experiments and finite element analysis of a scaffold architecture printed in PLA [12]. The mechanical properties of the PLA materials were good, and could guarantee good manufacturing accuracy. Therefore, it was very suitable as the manufacturing material for this model. The parameters are shown in Table 4.

Figure 20 and Figure 21 show the optimization results after the improved material interpolation. The results were postprocessed and the overall structure was clear. There was no fine treelike, difficult-to-print area, and it could be applied to additive manufacturing.

## 5. Conclusions

Based on the self-developed optimization-manufacturing integrated approach, this paper proposed an improved material interpolation sensitivity calculation method. The modeling comparison experiment on the space nodes verified that the integrated forming topology method had good feasibility. The conclusions are given below:
(1)The material interpolation method of the optimization algorithm was improved, the penalty effect of the material method was strengthened, and the adaptability was increased. Compared with the SIMP method, the secondary developed BESO method did not easily fall into the optimal local solution, and the performance of the obtained optimization results was better.(2)Based on the multisoftware collaborative computing platform independently developed by the research group and the model extraction method proposed in this paper, there was no need to modify the model grid and edit it again, so the method could easily and quickly realize the entire process of design-manufacturing.

This method combined the algorithm and the finite element analysis software in the form of a third-party plug-in, and completed the additive molding of the space node very well. In the future, its openness can provide a good framework platform for the addition of various constraints in the further refinement of topology technology. 

## Figures and Tables

**Figure 1 materials-15-03874-f001:**
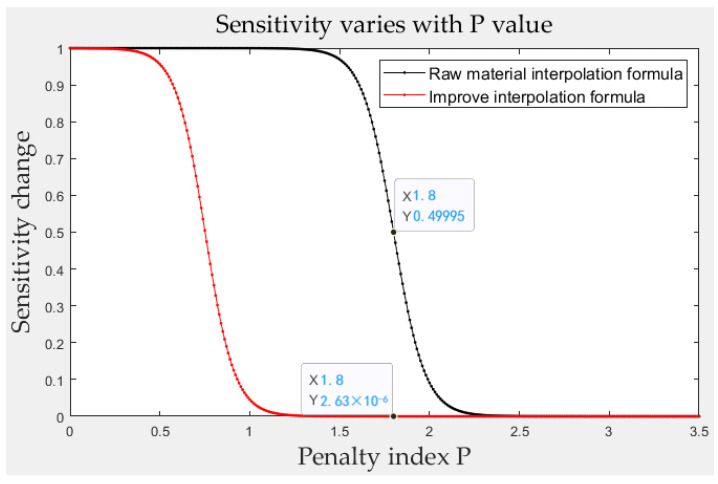
The variation in sensitivity with the penalty index *p* before the improvement in the material interpolation method formula (black) and after improvement (red).

**Figure 2 materials-15-03874-f002:**
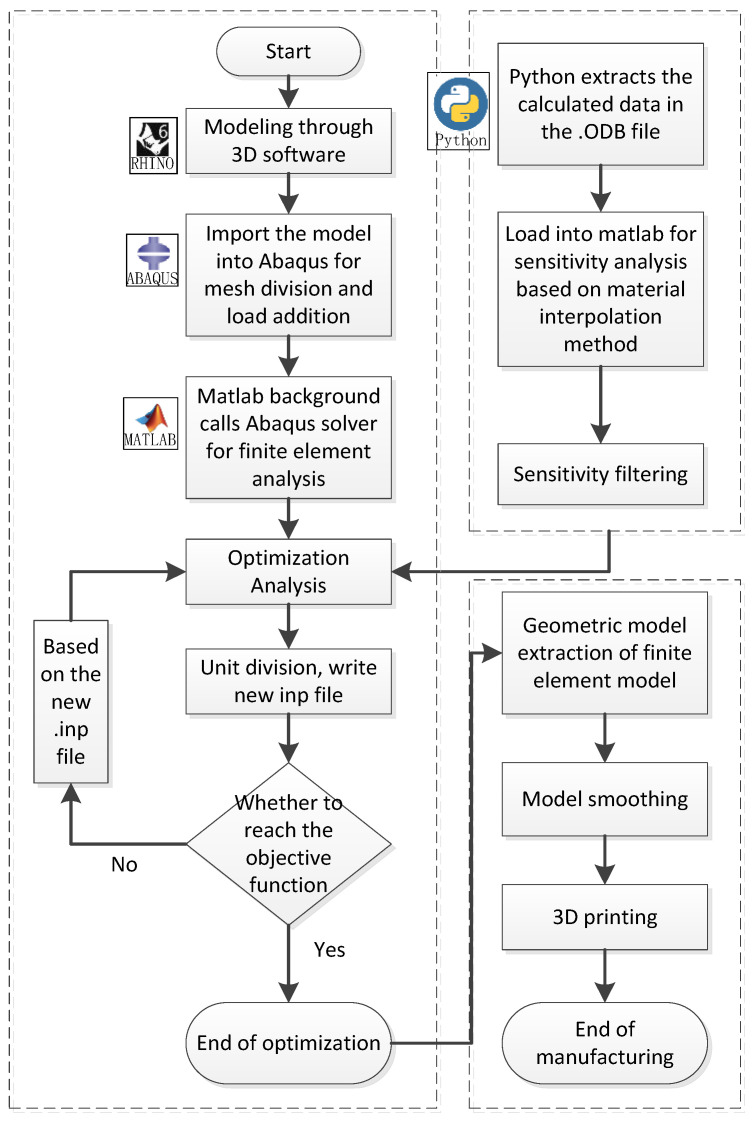
Optimization-manufacturing software integration method flow chart.

**Figure 3 materials-15-03874-f003:**
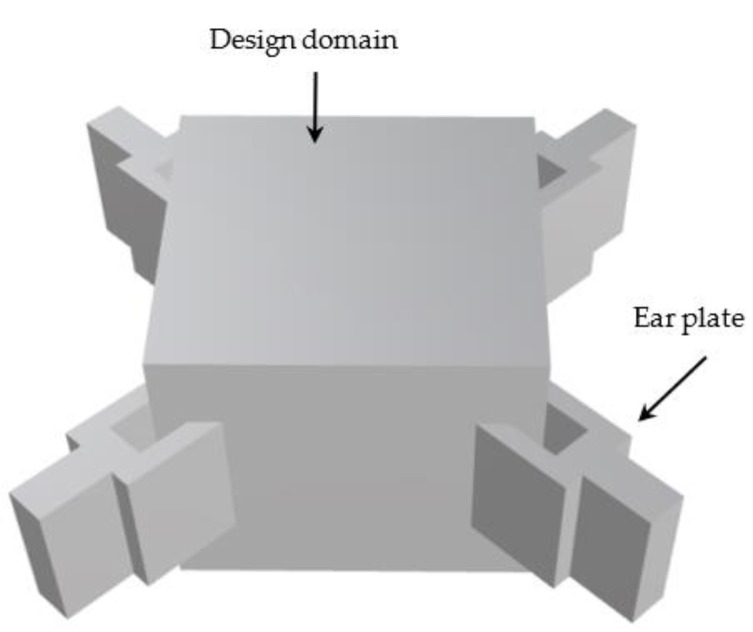
The node model established using the 3D design software Rhino, including the design domain and the part of the lug for applying the load.

**Figure 4 materials-15-03874-f004:**
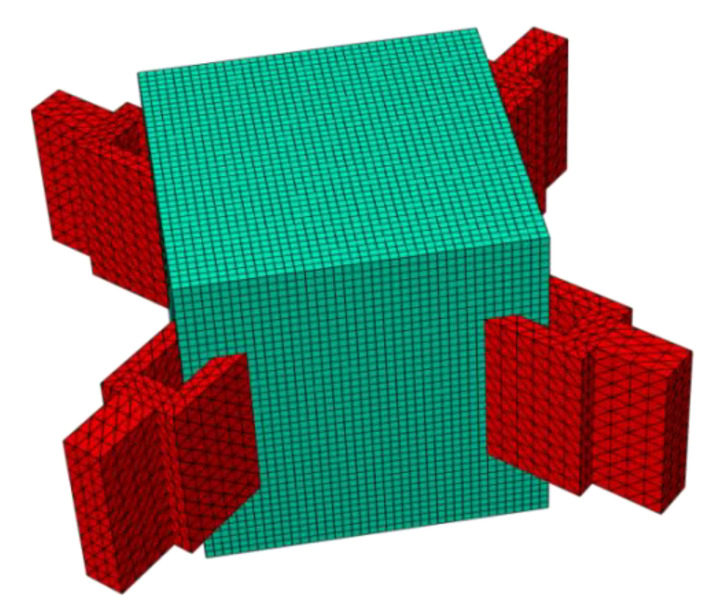
The node model of the finite element mesh division in ABAQUS (the red part is the load application area, and the green part is the design domain).

**Figure 5 materials-15-03874-f005:**
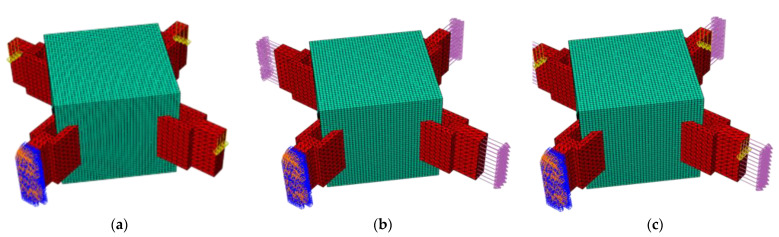
Schematic diagram of finite element application of different load cases (pink arrows are axial forces, yellow arrows are shear forces, orange and blue ends are where the boundary conditions are applied, and the green part is design domain): (**a**) shear load; (**b**) axial load; (**c**) combined load.

**Figure 6 materials-15-03874-f006:**
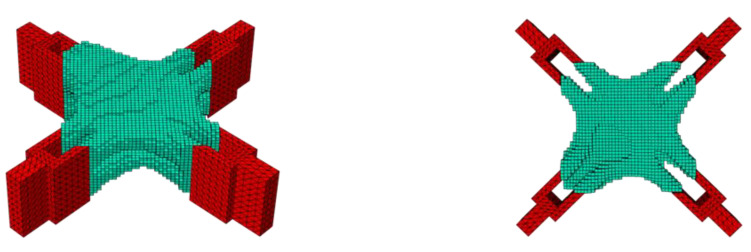
Optimized molding effect after improvement with different angles, showing a smoother profile (the red part is the load application area, and the green part is the design domain).

**Figure 7 materials-15-03874-f007:**
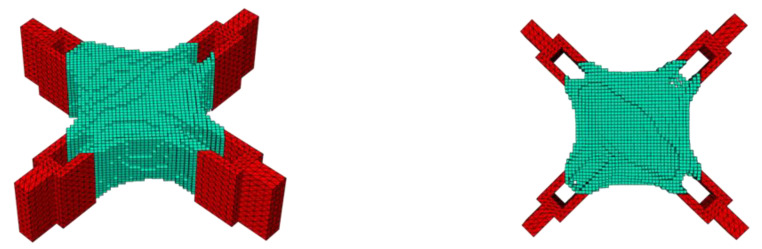
Optimized molding effect before improvement with different angles showing island units (the red part is the load application area, and the green part is the design domain).

**Figure 8 materials-15-03874-f008:**
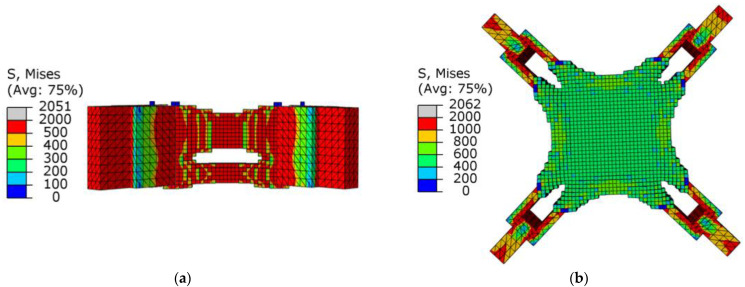
Schematic diagram of optimization results under axial load conditions: (**a**) cutaway view; (**b**) stress nephogram.

**Figure 9 materials-15-03874-f009:**
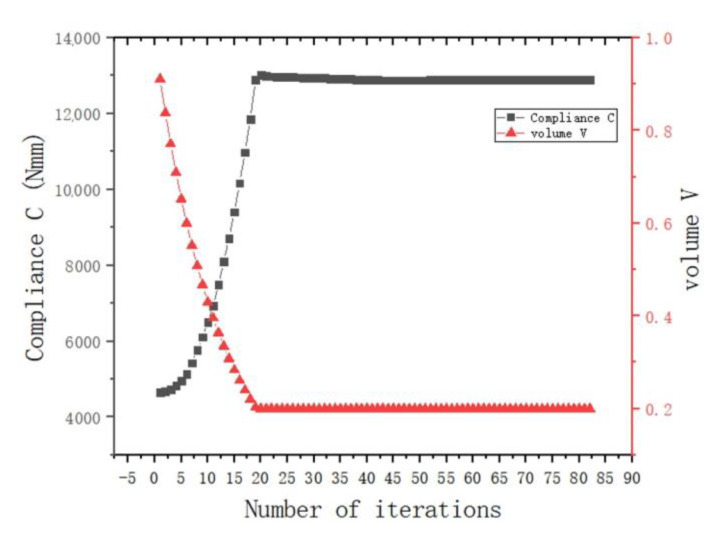
Optimization process under the condition of axial force (the black dotted line is the change curve of compliance C, and the red dotted line is the volume-reduction curve).

**Figure 10 materials-15-03874-f010:**
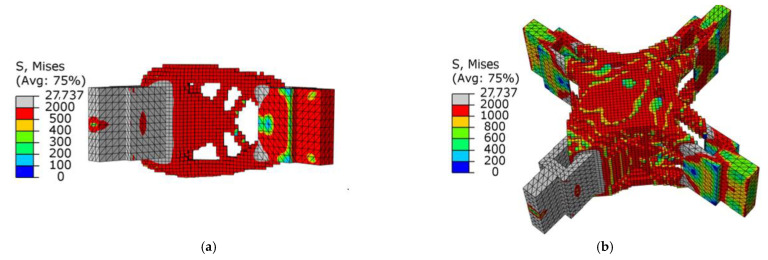
Schematic diagram of optimization results under shear conditions: (**a**) lateral view; (**b**) stress nephogram.

**Figure 11 materials-15-03874-f011:**
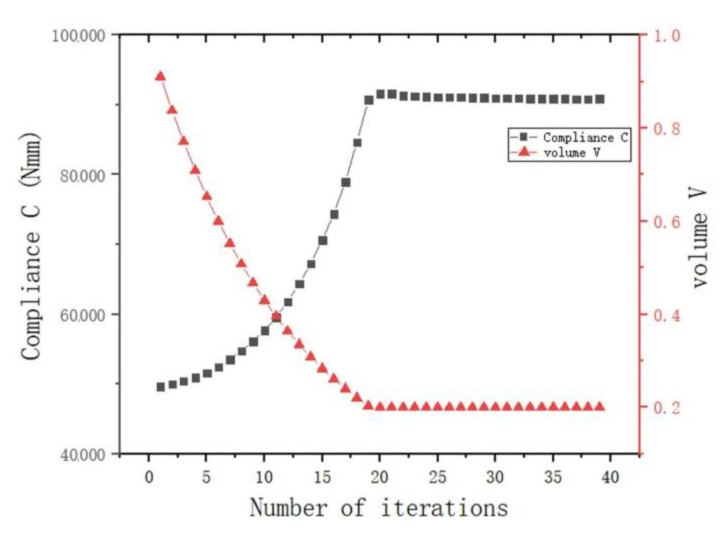
Optimization process under shearing conditions (the black dotted line is the change curve of compliance C, and the red dotted line is the volume-reduction curve).

**Figure 12 materials-15-03874-f012:**
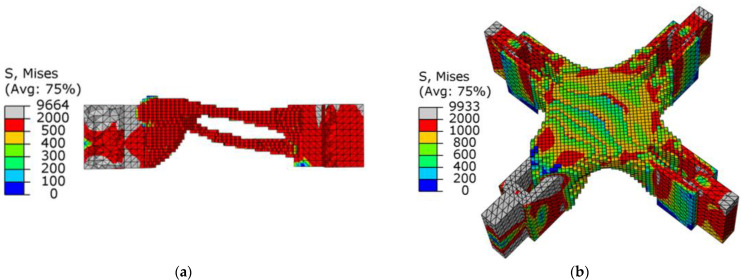
Schematic diagram of optimization results under combined conditions: (**a**) cutaway view; (**b**) stress nephogram.

**Figure 13 materials-15-03874-f013:**
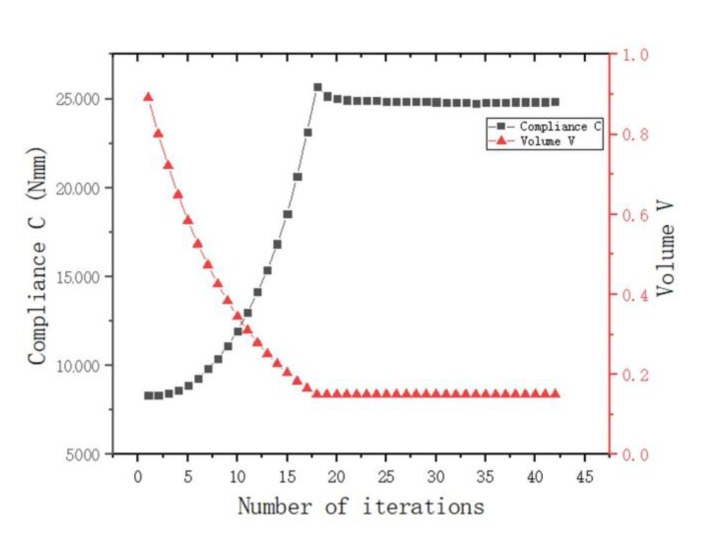
Optimization process under combined working conditions (the black dotted line is the change curve of compliance C, and the red dotted line is the volume-reduction curve).

**Figure 14 materials-15-03874-f014:**
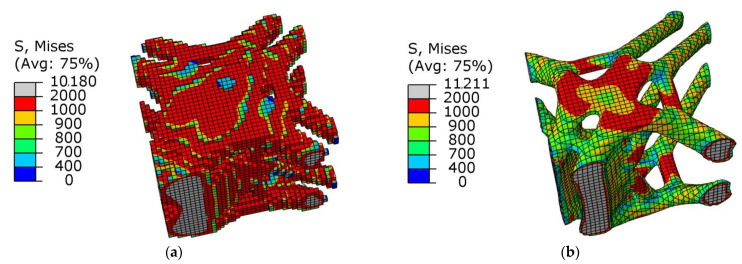
The stress nephograms of the design domain of the optimized joint under the shear condition: (**a**) BESO; (**b**) SIMP.

**Figure 15 materials-15-03874-f015:**
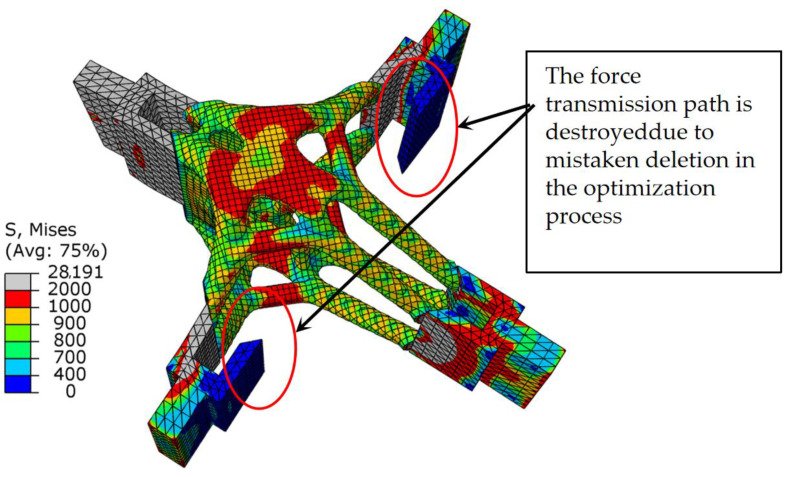
Stress nephogram of SIMP optimization results under shear conditions.

**Figure 16 materials-15-03874-f016:**
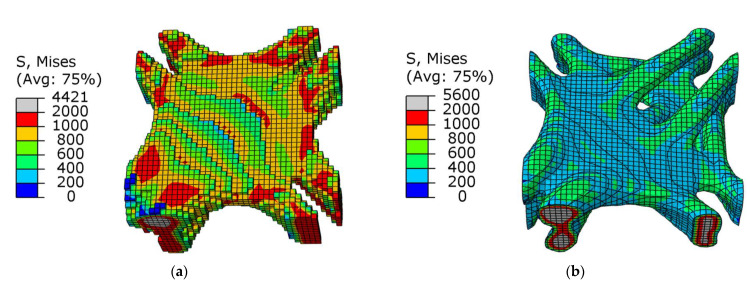
Stress cloud diagram in the design domain of optimized joints under combined conditions: (**a**) BESO; (**b**) SIMP.

**Figure 17 materials-15-03874-f017:**
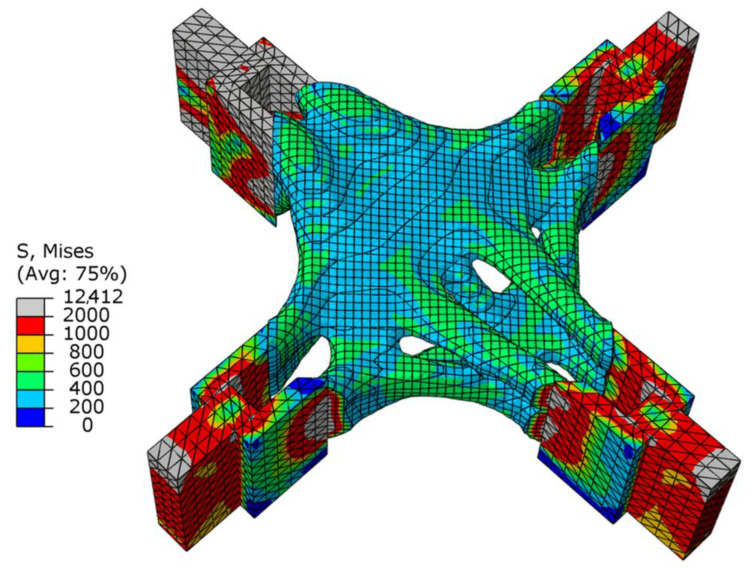
Stress cloud diagram of SIMP optimization results under shear conditions.

**Figure 18 materials-15-03874-f018:**
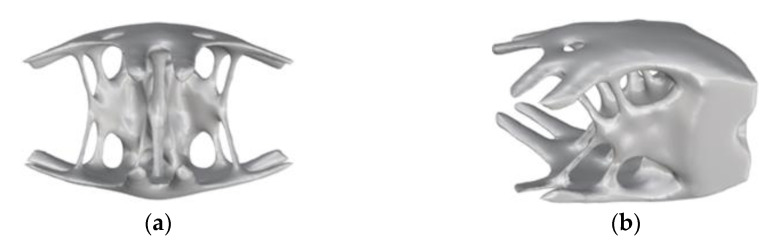
Smooth model of different angles under shear conditions: (**a**) front; (**b**) side.

**Figure 19 materials-15-03874-f019:**
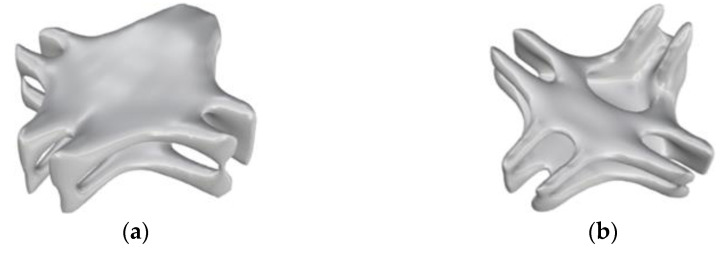
Smooth model of different angles under combined working conditions: (**a**) top; (**b**) bottom.

**Figure 20 materials-15-03874-f020:**
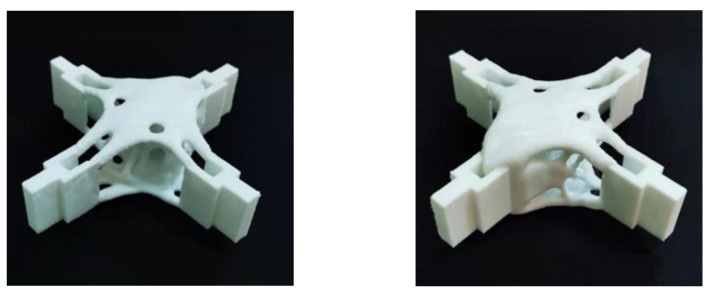
Print effect picture after improvement (Under shear conditions).

**Figure 21 materials-15-03874-f021:**
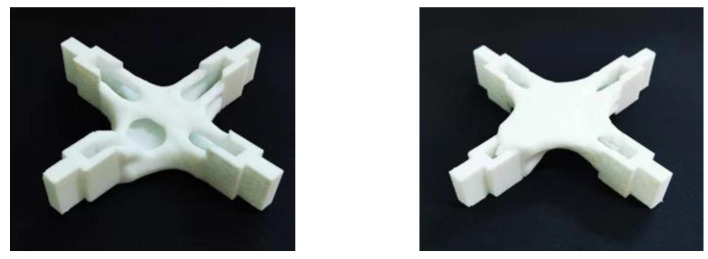
Print effect picture after improvement (combined working condition).

**Table 1 materials-15-03874-t001:** Structural parameters.

	Structure Size (mm)	Density (kg/m^3^)	Elastic Modulus (MPa)	Poisson’s Ratio
Design domain	80 × 80 × 80	7850	2.1 × 10^5^	0.28
Ear plate thickness	10	7850	2.1 × 10^5^	0.28
Ear plate height	40	7850	2.1 × 10^5^	0.28

**Table 2 materials-15-03874-t002:** Comparison of optimization results of shear conditions.

Algorithm	Maximum Stress (MPa)	Minimum Stress (MPa)
BESO	10,180	11.34
SIMP	11,210	4.1 × 10^−6^

**Table 3 materials-15-03874-t003:** Comparison of optimization results of combined working conditions.

Algorithm	Maximum Stress (MPa)	Minimum Stress (MPa)
BESO	4421	58
SIMP	5600	2.2 × 10^−6^

**Table 4 materials-15-03874-t004:** Process parameter table for the printing molding experiment.

Print Type	Material	Speed (mm/s)	Fill Rate	Temperature (°C)
FDM	PLA	60	15%	220

## Data Availability

Not applicable.

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
