# Peer review of "Research on 3D-Print Design Method of Spatial Node Topology Optimization Based on Improved Material Interpolation"

_materials, 2022, doi:10.3390/ma15113874_

Round 1
Reviewer 1 Report
I would like to appreciate the authors for conducting the research work entitled “Research on 3D-print Design Method of Spatial Node Topology Optimization Based on Improved Material
Interpolation”. Since, 3D printing products are nowadays more demandable and innovative manufacturing techniques seeking to be introduced in many industries across the countries. Hence conducting such a task would be very much helpful to the people/industries who expect seamless manufacturing. Therefore, it is quiet appreciable task in 3D printing. Based on the above task completed by the authors I would like to comment few points on the above work as a reviewer.
- What do you represent “Penalty Index”? Describe some more points therefore the readers can understand more on the specific term.
- How the soft cell sensitivity is reduced when penalty index raises.
- Is there any result were experimentally done for the FEA model?
- Most of the results are explored based on the FE analysis. Therefore, I would like to suggest the authors to compare the results with mechanical testing, that probably helps to the reader to understand the significance of the work.
Author Response
Dear reviewer, thank you very much for your valuable comments. I will reply to your questions point by point.
Point 1:What do you represent “Penalty Index”? Describe some more points therefore the readers can understand more on the specific term.
Response 1: Penalty Index: Material interpolation method derived from variable density method. This method changes the sensitivity of the finite element of the structure from only two representations of 0 and 1 to continuous [0, 1] because it is mathematically easier to optimize the solution for continuous variables than for discrete variables. Therefore, the concept of intermediate density is introduced so that the finite element has a value between 0 and 1. Otherwise, when there are only two numbers, 0 and 1, there are too many combinations to get the optimal solution. Generally, the interpolation method is to set the minimum value manually. Under the action of the penalty index, the sensitivity of the unit's location (soft unit) to be deleted infinitely approaches 0, that is, approximate deletion, because once it is equal to zero, the matrix will be singular. The penalty index is put into the material interpolation method formula as the index part, and its size can greatly affect the sensitivity of the soft element. (Added below Figure 1 in Section 2.2 of the article)
Point 2: How the soft cell sensitivity is reduced when penalty index raises.
Response 2: As the penalty index increases, the numerator of the material interpolation formula shrinks, and the result will shrink due to the presence of a fixed soft element elastic modulus as the denominator. Multiplying the calculated value of the soft element obtained by the finite element calculation to obtain the soft element's reduced sensitivity value, which gives practical meaning to the soft element as the deleted part in the Finite Element Analysis. In addition, in the following sensitivity filtering, the contribution of the soft unit sensitivity can also be reduced. (Added below Figure 1 in Section 2.2 of the article)
Point 3: Is there any result were experimentally done for the FEA model?
Response 3: NO. (Explained in the next question)
Point 3: Most of the results are explored based on the FE analysis. Therefore, I would like to suggest the authors to compare the results with mechanical testing, that probably helps to the reader to understand the significance of the work.
Response 4: The question you asked is critical. The model experiment is still under study. There are mainly the following concerns: Many articles have proved that 3D printed products have certain anisotropy problems, such as the weaker mechanical properties of 316L stainless steel products in the manufacturing direction. The team also conducted verification and found that 3D printing does indeed need to consider the anisotropy of the structure. The experimental model is necessary to optimize based on the anisotropic properties of the material to obtain an optimization model that conforms to the actual manufacturing to ensure the rigor of the design and experiment. So the experiment is still under study. This paper focuses on the secondary development of the optimization algorithm and introduces topology optimization and additive manufacturing into the research of space nodes to provide ideas for the optimal manufacturing of space nodes. After careful consideration of your suggestion, we have re-added in the article a comparison with the calculation results of the ABAQUS optimization module. ABAQUS, as a mature finite element software, uses the SIMP algorithm. A comparison is made at the algorithm level, and it is found that the algorithm used in this paper is indeed desirable(in section 3.4). I hope to get your approval. Regarding model experiments, we are more than willing to share with you after we have produced models that take into account anisotropy.

Reviewer 2 Report
In this work the authors studied a 3D-print design method optimization based on improved material interpolation. Manuscript is not well prepared and introduction, methodology, results/discussion sections need critical modifications; however, the following checks and comments can help improve the manuscript.
- The novelty of the work is not highlighted.
- Table and Figure’s captions need to be more informative.
- There are a lot of typos as well as grammatical issues. For example in Table 1, the dimension of Elastic Modulus is presented as “Mpa” and “structure size” is not capitalized, as well. Also, in page 11, line 307, “Table 1” must change to “Table 2”.
- In “Introduction”, just 8 papers were reviewed while at least half of them are old.
- Results and discussions has not compared to the findings of the other researchers.
- The authors are recommended to review the https://doi.org/10.1007/s11665-021-05894-y for the section “2 3D printing effect“.
Author Response
Dear reviewer, thank you very much for your valuable comments. At the same time, I am sorry for the various omissions in my article. Next, I will reply to your questions point by point.
Point 1: The novelty of the work is not highlighted.
Response 1: First, although both topology optimization and additive manufacturing technologies and theories are mature, structural design is often not so free due to various specifications in the construction field. In the current optimization design of spatial nodes, only a few studies use the BESO method because although the BESO theory has matured, a large number of optimization designs are carried out with more convenient software, and most of these software use the SIMP method. Of course, this also brings a problem, that is, the openness of the software is limited. Therefore, we think this paper has the following novelties:
(1).This paper develops the BESO method twice, and completes the entire process of optimizing manufacturing. While providing engineering application ideas, it can easily insert some manufacturing constraint algorithms according to requirements.
(2). We newly added the comparison of the results obtained by the BESO algorithm and the software SIMP algorithm, which shows the superiority of the calculation results of the BESO method in the results.
(3). The BESO method has less application and is related to the resulting rough boundary. Therefore, this paper uses the three-dimensional modeling software RHINO to explore the boundary smoothing method, which proves the achievability of the method.
Novelty has been re-emphasized in the abstractFirst, although both topology optimization and additive manufacturing technologies and theories are mature, structural design is often not so free due to various specifications in the construction field. In the current optimization design of spatial nodes, only a few studies use the BESO method because although the BESO theory has matured, many optimization designs are carried out with more convenient software, and most of this software uses the SIMP method. Of course, this also brings a problem: the openness of the software is limited. Therefore, we think this paper has the following novelties:
(1). This paper develops the BESO method twice and completes the entire process of optimizing manufacturing. While providing engineering application ideas, it can easily insert some manufacturing constraint algorithms according to requirements.
(2). We added the comparison of the results obtained by the BESO algorithm and the software SIMP algorithm, which shows the superiority of the calculation results of the BESO method in the results.
(3). The BESO method has less application related to the resulting rough boundary. Therefore, this paper uses the three-dimensional modeling software RHINO to explore the boundary smoothing method, which proves the achievability.
I have re-emphasized novelty in the abstract.
Point 2: Table and Figure’s captions need to be more informative.
Response 2: Information on tables and pictures has been added.
Point 3: There are a lot of typos as well as grammatical issues. For example in Table 1, the dimension of Elastic Modulus is presented as “Mpa” and “structure size” is not capitalized, as well. Also, in page 11, line 307, “Table 1” must change to “Table 2”.
Response 3: Thank you for your careful review. Spelling and grammar errors have been corrected.
Point 4: In “Introduction”, just 8 papers were reviewed while at least half of them are old.
Response 4: The introduction has been reworked.
Point 5: Results and discussions has not compared to the findings of the other researchers.
Response 5: The comparison with the calculation results of the commercial finite element software SIMP algorithm has been re-added in the article.
Point 6: The authors are recommended to review the https://doi.org/10.1007/s11665-021-05894-y for the section “2 3D printing effect“.
Response 6: Thank you for this article. I am very interested in this. Many articles on additive manufacturing also point out that additive manufacturing has a certain anisotropy, with different mechanical properties in the stacking direction and the scanning path direction. Structural form, interlayer fusion quality, scanning speed, energy, contact area, etc., will affect the overall performance of 3D printing. This paper only considers the molding effect for the time being. It focuses on the secondary development of the optimization algorithm, introducing topology optimization and additive manufacturing into the study of space nodes, and providing a convenient method for the optimal manufacturing of space nodes. Our research group is already working on researching various properties of additively manufactured products.
We very much want your approval.

Round 2
Reviewer 1 Report
All the necessary corrections are made in the manuscript as per the comments. Therefore, the manuscript can be accepted in the present form.
Author Response
Dear reviewer, thank you very much for your patient review!
Reviewer 2 Report
Most of the comments were not addressed, so I can not accept it.
My unanswered comments are listed below:
- Captions of the Figures 1,3,4,5,9,11,13 need to be more informative, the authors did not care to reach out.
- The authors should highlight the corrections for tracking! I do not know where and what they did corrections.
- In “Introduction”, the number of the papers that were reviewed is still insufficient and old.
- In the Results and Discussions section, the authors did not clearly point out the changes.
- The authors did not review the recommended paper; https://doi.org/10.1007/s11665-021-05894-y for the section “4.2 3D printing effect“.
Author Response
Dear reviewer, thank you very much for your patient reply. I have carefully checked the relevant articles and made the following revisions. Please give me another chance.
- The title information has been supplemented under the specified image and highlighted.
- All changes have been highlighted.
- In the introduction, related research in this field is added, the number of papers reviewed has been increased to 17, and the papers reviewed are relatively recent and classic.
- A comparison with the Tosca module of ABAQUS is added in the article. In the finite element analysis, it can be seen that the forming effect of this algorithm is better, it is not easy to fall into the optimal local solution, and the performance of the optimization result is also better. It has been added to the conclusion and highlighted.
- The authors reviewed the article recommended by the reviewer, which have been highlighted in Section 4.2. The recommended article looks at the experiments and finite element analysis of additive manufacturing, and I am admired that such a highly accurate experiment can be carried out with such a small rod.

Round 3
Reviewer 2 Report
Overall, the authors provided proper answers to the query of the reviewer and adjusted the manuscript. In this circumstance, I recommend acceptance of the current version.